# Oral Health Professionals' and Patients' Opinions of Type-2 Diabetes Screenings in an Oral Healthcare Setting

Rodrigo Mariño *[iD], Andre Priede, Michelle King [iD], Geoffrey G. Adams, Maria Sicari and Mike Morgan

Melbourne Dental School, University of Melbourne, Melbourne 3010, Australia
* Correspondence: r.marino@unimelb.edu.au; Tel.: +61-437-808-554

**Abstract:** Objectives: As part of an evaluation of an oral healthcare practice-based model that identifies patients with prediabetes or type-2 diabetes (T2D), this study reports on the experiences and opinions of oral health professionals and patients on the screening program. Methodology: Urban and rural oral healthcare practices were invited to participate. Participating practices invited eligible patients to participate in the screening program. Patients were categorised as low, intermediate, or high-risk for prediabetes/T2D. Patients in the intermediate or high-risk category were referred to their general practitioner (GP) for further investigation. Post-screening surveys were used to assess acceptability, barriers and facilitators of the screening program among participating oral health professionals (OHP) and patients. Results: The post-screening survey was completed by 135 patient, and 38 OHPs (i.e., dentists, dental hygienists, oral health therapists). the majority of OHPs (94.6%) who delivered the protocol were satisfied with the approach. Most patients reported satisfaction with the approach (73.2%) and would recommend it to others. Several barriers for implementation were identified by OHPs and patients. Conclusion: OHPs feedback indicated that the screening model was generally acceptable. The feedback from patients following their participation in this study was overwhelmingly positive, indicating that the screening protocols were accepted by patients.

**Keywords:** general practitioners; oral health professionals; screening; type-2 diabetes





## 1. Introduction

Early diagnosis and treatment of type-2 diabetes (T2D) is critical to improving health outcomes. Timely interventions lead to the prevention of progression of this condition and a reduction in the risk of associated complications, such as strokes, retinopathy, nephropathy, heart failure, amputations, and feet ulcers [1,2]. Diabetes is also relevant to oral health as oral health patients with poorly controlled diabetes experience far greater periodontal problems and poorer treatment outcomes, eventually leading to tooth loss, compared to those who keep their blood sugar within normal limits [3]. Thus, oral health professionals (OHP) can have an important role in T2D prevention and identification. For some patients, a dental visit may be their only point of contact with the health care system [4]. Such a screening would contribute to the early detection of cases of T2D. Additionally, it would increase awareness and engagement in T2D prevention for those at higher risk. Reports have demonstrated that an individual is more likely to modify his/her lifestyle when they are aware of his/her prediabetes condition [5].

An oral healthcare practice-based model that identified patients with prediabetes or T2D and barriers to the implementation of the model (the iDENTify study) was tested in the state of Victoria, Australia [6]. The field component of the iDENTify study indicated that of the patients whose results were returned to OHPs (n = 96; 25.0%), six (6.3%) had undiagnosed prediabetes/T2D, which is within the range reported in the literature [7]. More importantly, those findings confirmed that identifying individuals at high-risk of having or developing T2D is effective, feasible, and could be part of a routine oral health-care assessment.

However, an outcome evaluation provides only limited information about OHPs and patients' experiences and perceptions of factors contributing to the successful implementation of screening programs [8,9]. For iDENTify to be successful, it is essential that there are close collaborations between OHPs, patients, and GPs. This collaboration may require additional OHP education and professional development to make sure that those at higher risk of T2D are referred for further assessment.

To obtain a more comprehensive understanding of the approach to identifying patients with prediabetes or T2D, an evaluation was undertaken to investigate OHPs and participating patients' opinions on the screening program as well as facilitators and deterrents to this practice-based model adoption. This information will be used to obtain a better understanding of the acceptability of this approach among OHPs and patients, and subsequently to inform the development of continuing education programs specifically focused on T2D/prediabetes prevention, identification, and management.

## 2. Material and Methods

Formal ethical approval was obtained from the University of Melbourne Human Research Ethics Committee. All research tasks were performed following the approved methodology and in accordance with the relevant guidelines and regulations. All participants provided informed consent to participate in this study. With that approval, tailored completion surveys were provided for each consenting OHP and patient participant. Surveys were provided either by email or post to OHPs via a link to a Qualtrics survey (Qualtrics, Provo, UT) to complete if an email address was provided at recruitment. Patient who indicated at recruitment that they would prefer to be posted the completion survey were mailed a hard copy with a reply-paid envelope. Reminder e-mails were sent to all OHPs and patients. Data collection for this component of the study commenced in September 2018 and was completed in June 2020, at the end of the data collection period.

The iDENTify screening protocol consists of several phases, starting with patient engagement and finishing with the OHP receiving the medical diagnosis for the patients they had referred to the GP. This approach is fully described elsewhere [10]; however, put briefly, interested participants were provided with a pack that included the plain language statement, patient consent form, The Australian diabetes risk assessment tool (AUSDRISK) and information about T2D and oral health. The AUSDRISK consists of a series of items, including age, sex, ethnic background, family history of diabetes, history of high blood glucose, hypertension, smoking history and status, nutritional information, physical activity, and waist measurement [11]. The AUSDRISK score classifies individuals as being at 'Low risk' ($\leq$5); 'Intermediate risk' (6–11); or 'High risk' ($\geq$12) of developing T2D. Those who had an AUSDRISK score of 6 or above were considered to be at high-risk and invited to participate in the iDENTify study. The OHP then provided a referral letter and information pack for the patient to take to their general medical practitioner (GP).

On completing the oral examination and referral of intermediate and high-risk patients, OHPs were asked to complete a six-item assessment questionnaire to assess their views on the usefulness of the approach, experience, and perceptions about the experience. The completion survey consisted of statements that participants rated on a five 5-points Likert scale, depicting their level of agreement with the six statements. Participants were asked to self-assess their level of satisfaction with the iDENTify approach, with the additional tasks involved with this approach, and the GP's review of referred cases utilizing a 5-points Likert scale (1 = 'Strongly satisfied'; 3 'Neutral'; 5 'Strongly dissatisfied'). The clarity of instructions and the level of difficulty in getting responses from the GP were also assessed in a five 5-points Likert scale (1 'Very easy'; 3 'Neutral'; 5 'Very difficult'). Participants were also asked about their level of recommendation for the iDENTify approach to be implemented in an oral healthcare setting from 1 = 'Would recommend'; 3 'Neutral'; 5 'Would not recommend'.

As a further verification of the approach, the length of the interaction between oral health professional and the patient was evaluated [Was the initial oral health examination

too long (risk assessment, periodontal exam, and referral)?] utilising a three-option response of 'Yes', 'No' or, 'Neutral'.

The completion evaluation also contained three open-questions, so participants could include their thoughts about their experience and their appraisals. OHPs and patients had the opportunity to expand on their answers. Additionally, OHPs had the opportunity to provide a reason as to why a participant was not referred.

The OHPs' survey data also included sociodemographic and work characteristics (5 items). The sociodemographic information included, sex, age-range, professional group (dentists; dental hygienists; dental therapists and oral health therapists), professional experience and location of practice.

The completion form for patients included nine items. As with OHPs, items included their level of satisfaction with the approach, level of difficulty in understanding procedures, level of satisfaction with the procedures, and level of recommendation of the approach. Additionally, the patients' completion form consisted of four items that participants rated on a five 5-points Likert scale, including level of difficulty to complete the type-2 diabetes risk assessment tool (i.e., AUSDRISK), the level of satisfaction with the oral examination, and level of agreement with a statement about their level of awareness before and after participating in the iDENTify project (1 'Strongly agree'; 3 'Neutral'; 5 'Strongly disagree'). Patients were also asked regarding their perceived level of difficulty in booking an appointment with the GP. Originally, it was planned to mail a completion survey to GPs in March 2020. Due to COVID-19 and increased pressures on general practice, it was deemed inappropriate to send the survey.

*Data Analysis*

Due to the small sample size, only descriptive analysis was used to illustrate the participants' views about the format, content, and delivery of the iDENTify approach. In some cases, Chi square analysis was used to compare results between responses and the distribution of sociodemographic. Data manipulation and analyses were conducted using IBM SPSS Statistics (Version 23.0, IBM Corporation, Endicott, NY, USA). In addition to the descriptive data, this analysis provides insight into the OHP and patients' satisfaction with the iDENTify approach.

## 3. Results

### 3.1. Oral Health Professionals (OHP) Completion Survey Response

From a total of 76 OHPs participated participating in the iDENTify project, 38 OHPs provided feedback on our approach for the identification of T2D and prediabetes in an oral healthcare setting (response rate 50%). One response was incomplete and was excluded from the analysis. Of the remaining 37 OHPs (31 dentists; 2 dental hygienists; and 4 oral health therapists), the majority of OHPs were either strongly satisfied or satisfied (94.6%) with the iDENTify approach and "would recommend" would "likely recommend" its implementation in oral healthcare settings (81.0%) (See Table 1). OHPs indicated advantages of the approach that went beyond the identification of T2D patients such as promoting better teamwork and "a better relationship between GP and dentists in managing a patient". On the other hand, one OHP reported they would likely not recommend its implementation. No reason was given for that opinion.

The majority of OHPs (69.4%) did not consider the oral examination (risk assessment, periodontal exam, and referral) too long. OHPs were also generally satisfied with completing the additional tasks involved with this approach (88.9%). One OHP was slightly dissatisfied, but the reason was related to research driven activities rather than the clinical procedures (e.g., "It took a fair bit of time to explain to patients and conduct the survey procedure").

**Table 1.** Oral health professionals completion survey rResponses (%).

| 1. How satisfied were you with the iDENTify? [a] | | | | |
|---|---|---|---|---|
| Strongly satisfied | Slightly satisfied | Neutral | Slightly dissatisfied | Strongly dissatisfied |
| 67.6 | 27.0 | 5.4 | – | – |
| 2. If this approach for detection of type-2 diabetes were available for patients, how likely would you be to recommend its implementation in an oral healthcare setting? | | | | |
| Would recommend | Likely recommend | Neutral | Likely not recommend | Would not recommend |
| 40.5 | 40.5 | 16.3 | 2.7 | – |
| 3. How satisfied were you with completing the additional tasks (AUSDRISK, periodontal exam, referral) as part of your oral health check-up? | | | | |
| Strongly satisfied | Slightly satisfied | Neutral | Slightly dissatisfied | Strongly dissatisfied |
| 61.1 | 27.8 | 8.3 | 2.8 | – |
| 4. Were instructions and information about iDENTify easy to understand? | | | | |
| Very easy | Slightly easy | Neutral | Slightly difficult | Very difficult |
| 73.0 | 16.2 | 10.8 | – | – |
| 5. Was it difficult to get a response from the General Medical Practitioner? | | | | |
| Very easy | Slightly easy | Neutral | Slightly difficult | Very difficult |
| 5.7 | 2.9 | 45.6 | 22.9 | 22.9 |
| 6. How satisfied are you with the review of your prediabetes and type-2 diabetes by the General Medical Practitioner? | | | | |
| Strongly satisfied | Slightly satisfied | Neutral | Slightly dissatisfied | Strongly dissatisfied |
| 2.8 | 28.6 | 45.8 | 17.1 | 5.7 |

[a] *n* = 37.

When OHPs were asked about the level of difficulty with following instructions, the majority found it very easy (73.0%) or slightly easy (16.2%). Another 10.8% of them was neutral.

However, when OHPs were asked about the level of difficulty getting a response from the GP, the largest group found it either very difficult or slightly difficult (45.8%) and another 45.6% were neutral about the level of difficulty. However, it is difficult to assess the real reason for this difficulty. One OHP indicated failure to receive a response may not have been due to the GP but instead due the patient not attending medical follow-up:

> *"I think we did not get a response from GP because the patients did not consult their GPs but for those who did consult, their GPs responded."*

Although about one-third of OHPs (31.4%) was satisfied with the review of cases by the GP, the largest group of participants (45.8%) was neutral about their level of satisfaction with the review of their prediabetes and T2D status by the GP. The remainder (23.9%) was either slightly dissatisfied or strongly dissatisfied. Furthermore, one OHP called for the strengthening of the communications between OHPs and medical practitioners:

> *"Extremely important study in bridging the gap between dentist and medical personnel."*

*3.2. Patient Completion Survey Responses*

At the end of data collection, 135 patients from the 384 referred to their GP for T2D assessment and provided feedback on the iDENTify approach for the identification of T2D and prediabetes in oral healthcare settings, achieving a response rate of 35.1%.

The majority of patients were either "Very satisfied" or "satisfied" (73.2%) with the iDENTify approach and would 'strongly recommend' or 'recommend' the practice to other people (85.0%) (see Table 2). Furthermore, one participant called for an expansion of the iDENTify approach to other chronic conditions ("I like the wholistic aspects between separate but related disciplines. Should be more of it. e.g., relationship between gum disease and heart disease.").

**Table 2.** Oral Health Patient Completion Survey Reponses (%).

| 1. How satisfied were you with the iDENTify? [a] | | | | |
|---|---|---|---|---|
| Strongly satisfied | Slightly satisfied | Neutral | Slightly dissatisfied | Strongly dissatisfied |
| 56.0 | 17.2 | 24.6 | 2.2 | – |
| 2. If this approach for detection of diabetes was available for patients as standard practice, would you recommend it to other people? | | | | |
| Strongly recommend | Slightly recommend | Neutral | Slightly not recommend | Strongly not recommend |
| 55.6 | 29.4 | 13.4 | 1.6 | – |
| 3. After participating in this study I am more aware of prediabetes and type-2 diabetes. | | | | |
| Strongly agree | Slightly agree | Neutral | Slightly disagree | Strongly disagree |
| 51.5 | 25.8 | 18.8 | 1.6 | 2.3 |
| 4. Were instructions and information about iDENTify easy to understand? | | | | |
| Very easy | Slightly easy | Neutral | Slightly difficult | Very difficult |
| 76.3 | 13.0 | 10.7 | – | – |
| 5. How satisfied were you with completing the additional tasks (AUSDRISK, periodontal exam, referral) as part of your oral health check-up? | | | | |
| Strongly satisfied | Slightly satisfied | Neutral | Slightly dissatisfied | Strongly dissatisfied |
| 59.3 | 13.3 | 25.8 | 1.6 | – |
| 6. How difficult was the AUSDRISK assessment to complete? | | | | |
| Very easy | Slightly easy | Neutral | Slightly difficult | Very difficult |
| 73.0 | 13.5 | 11.9 | 1.6 | – |
| 7. How satisfied are you with the initial oral health examination including the periodontal examination? | | | | |
| Strongly satisfied | Slightly satisfied | Neutral | Slightly dissatisfied | Strongly dissatisfied |
| 79.0 | 12.4 | 7.8 | 0.8 | – |
| 8. Was it difficult to book an appointment with your General Medical Practitioner? | | | | |
| Very easy | Slightly easy | Neutral | Slightly difficult | Very difficult |
| 44.7 | 3.8 | 45.7 | 4.8 | 1.0 |
| 9. How satisfied are you with the review of your prediabetes and type-2 diabetes by your General Medical Practitioner? | | | | |
| Strongly satisfied | Slightly satisfied | Neutral | Slightly dissatisfied | Strongly dissatisfied |
| 41.5 | 9.4 | 49.1 | – | – |

[a] *n* = 135.

More importantly, when asked whether participating in the study increased their awareness about T2D, most patients either strongly agreed (51.5%) or slightly agreed (25.8%), and another 18.8% was neutral. One participant commented:

> *"Because of my lifestyle and diet, I had not considered myself to be at risk of type-2 diabetes. However, until being invited to partake in this study, I had not been aware that prediabetes could be detected through oral/dental examination".*

On the other hand, as expected, the iDENTify approach was less useful for those who had recently been assessed for prediabetes and T2D by their GP. Still, participants could appreciate the value of broadening diabetes screening into the oral healthcare setting, as a valuable tool that could reach more individuals:

> *"I have a regular check-up with my doctor for diabetes as well as other things so this not directly relevant to me, but I see it having big implications for people who are not otherwise being screened."*

When asked about the clarity and comprehensibility of the material covered, there was general agreement that the m aterial presented was clear and relevant to the purposes of this project. Most patients (89.3%) found it "very easy"/"easy" to understand the instructions and information received about iDENTify. In fact, one participant mentioned:

> *"I am happy to keep my teeth and gums in good condition but hadn't thought about prediabetes check until I met the iDENTify team."*

When patients were asked about how satisfied they were with completing the patient tasks as part of the approach, the majority was either strongly satisfied (59.3%) or slightly satisfied (13.3%), and one quarter (25.8%) were neutral. In the same way, the majority

also considered completing the patient tasks to be either very easy (73.0%) or slightly easy (13.5%). Another 11.9% were neutral about the degree of difficulty. Furthermore, one patient indicated that they had forgotten about their involvement in the project, thus suggesting that the tasks may not have been time-consuming or difficult for the patients to perform; they amount to short term, common activities of low impact, which are more likely to be underreported or forgotten by respondents [12].

The majority reported their level of satisfaction with the dental examination to be either strongly satisfied (79.0%) or slightly satisfied (12.4%). On the other hand, ten participants (7.8%) were neutral about the dental examination, and one was slightly dissatisfied. Asked about the level of difficulty in booking an appointment with GP, the largest group of participants was neutral (45.7%) and another 5.8% found it difficult or very difficult. In other words, less than half of respondents (48.5%) agreed that booking of medical appointments was generally easy. One patient feedback was:

> "I have been unable to follow up with a GP on a number of health checks due to the COVID-19 restrictions, so the timing was unfortunate."

In the same manner, almost half of the participants were neutral about their level of satisfaction with the review of prediabetes and T2D by their GP (49.1%). The remainder, 50.1% were either satisfied or strongly satisfied.

## 4. Discussion

Successful implementation of the proposed diabetes screening model requires the support of the oral health care professionals who deliver the protocol, patients, and GPs. In the iDENTify project, the majority of OHPs (94.6%) who delivered the protocol were satisfied with the approach, considered the additional tasks required for diabetes screening easy to perform, i.e., not excessively time consuming, and would recommend the protocols implementation in the oral healthcare setting. OHPs also considered that the procedures were easy to follow and could be incorporated into a standard practice.

Results from international studies generally correspond with present findings, with most OHPs reporting a favourable experience with diabetes screening in the oral healthcare setting. A UK study that explored oral healthcare staffs' experience with a diabetes screening programme in dental practice settings found the protocol easy to follow and implement, enabled staff to provide brief advice based on the risk factors, highlighted the links between general and oral health, and enabled them to communicate this association to patients [13]. The training staff undertook as part of the screening programme improved staff knowledge and understanding of T2D and confidence about implementing a diabetes risk score tool [13].

Qualitative studies have shown that diabetes uptake may be influenced by the screening methodology, with less invasive and time-consuming methods, such as those implemented in this study, being associated with an increased patient uptake [14,15]. Interestingly, barriers named in the literature [8,16–18] and others cited by OHPs in this project [19], were not raised by OHPs who participated in the clinical stage of the iDENTify project.

The value and effectiveness of any screening strategy is dependent on successful patient uptake, and a key determinant of participation is patients finding the protocol acceptable [14]. The feedback from patients following their participation in this study was overwhelmingly positive, indicating that the screening protocols were accepted by patients. The majority of participating patients reported satisfaction with the approach (73.2%) and would recommend it to others. Patients also found instructions easy to understand and considered the tasks required of them, to be easy to perform.

These results are comparable to those reported in international studies [17,20–22] that found the majority (72–90%) of patients were willing to be screened for diabetes in a private oral healthcare setting. These studies reported facilitators to patient participation including no additional financial costs incurred, well trained staff, confidentiality, a follow-up assessment with a GP was arranged by the staff, and a previously existence of a good rapport with the dentist [23].

Apart from the early identification of asymptomatic individuals, additional benefits from screening programmes have included participants valuing the information provided about the bidirectional relationship between diabetes and periodontal disease [20] and an improved perception of the dentists' professionalism, compassion, competence, and knowledge by participants [21]. Importantly, most patients agreed that participation in the programme increased their awareness about T2D. Consequently, as well as identifying individuals with prediabetes, others who completed the T2D risk assessment may have benefitted from raised understanding and awareness of prediabetes and T2D, and the modifiable risks for developing T2D.

When considering the result of this evaluation, the limitations of the iDENTify approach should be taken into account. The completion rates for patients, and to some extent for OHPs, was lower than expected. Although this study achieved good representation across oral health professionals and patients, further research may be needed to understand the reasons for this and develop strategies to overcome it. Additionally, dental practices, OHPs, and patients consisted of volunteers, which may have resulted in a positive bias in findings. In addition, the established OHP–patient rapport has limitations as well as benefits. For example, although responses were anonymous and independent, the desire to please the practitioner may affect participant patients' responses; moreover, the disclosure of alternative views may be less likely to occur.

For the iDENTify protocol to be effective, co-operation is required between OHPs and GPs. Therefore, referral pathways with GPs are essential for the effective implementation of the screening protocol. One potential barrier to completing the iDENTify protocol highlighted by the screened patients was arranging a follow-up appointment with a GP. Less than half of respondents reported that the booking of medical appointments was easy, and only half of the participants were satisfied or strongly satisfied with the review undertaken by the GP. However, the COVID-19 pandemic may have impacted on patients' willingness to attend their GP for further T2D assessment and on OHPs in collecting result.

Another limitation of our study was that feedback from GPs was not sought. Nonetheless, Australian studies that have explored GPs' recruitment and involvement in primary care research have highlighted potential barriers to GP involvement, including time constraints, workforce shortages, lack of remuneration and infrastructure support, and lack of interest in the research question [24]. On the other hand, GPs, dentists, and patients involved in a study where GPs were trained to identify dry mouth in a cohort of their older patients, gave very positive feedback about the suitability and acceptability of the model [24,25]. This collaboration continued in that project, greatly facilitating the development of pathways of care between OHPs and GPs [24,25]. Therefore, additional research is required to identify barriers and facilitators to GPs communicating the results of diabetes screenings to the OHP, as well as to better understand GPs opinions regarding OHPs screening for diabetes in the oral healthcare setting, so as to maximise the effectiveness of routine diabetes screening in the oral health care setting, or that of screening, GP referral, and the management of other chronic diseases in those settings.

The present experience highlights the importance and feasibility of utilising oral healthcare settings to screen for T2D to enable the identification of undiagnosed T2D/prediabetes cases. However, to generate strong and lasting effects, additional and continued activities may be needed. Further research should be conducted to better determine barriers and facilitators for OHPs referring individuals with an elevated risk of T2D to their GPs for further investigation. Additional research is also required to identify those barriers and facilitators to the screening of other chronic diseases in the oral healthcare setting including screening, GP referral and management. The sustainability of this approach is another important outcome to be explore. Engaging stakeholders in the process might also require that resources are invested, and that financial aspects are addressed under a T2D prevention perspective to facilitate changes, which can only be achieved in the long term.

**Author Contributions:** R.M.: Participated in the conception and design of the study, acquisition of data, analysis, and interpretation of data; as well as drafting of the manuscript and its critical revision, and approval of the final version. A.P.: Participated in the conception and design of the study, acquisition of data, analysis, and interpretation of data; as well as critical revision, and approval of the final version. M.K.: Participated in the design of the study, acquisition of data, analysis and interpretation of data; as well as critical revision, and approval of the final version. G.G.A.: Participated in the conception and design of the study, analysis and interpretation of data; as well as critical revision of manuscript, and approval of the final version. M.S.: Participated in the acquisition of data, as well as critical revision, and approval of the final version. M.M.: Participated in the conception and design of the study, analysis, and interpretation of data; as well as critical revision, and approval of the final version. All authors have read and agreed to the published version of the manuscript.

**Funding:** This project was funded by Colgate-Palmolive Pty Limited the provider of an unrestricted research grant to facilitate this research project. The authors would also like to acknowledge the support from the EviDent Foundation.

**Institutional Review Board Statement:** The study was conducted in accordance with the Declaration of Helsinki, and approved by the Institutional Review Board (or Ethics Committee) University of Melbourne Human Research Ethics Committee (protocol code ID: 1749595; 6 June2019).

**Informed Consent Statement:** Informed consent was obtained from all subjects involved in the study.

**Data Availability Statement:** The datasets generated and/or analysed during the current study are not publicly available due to the ethics approval granted on the basis that only researchers involved in the study could access the de-identified data. The minimum retention period is 5 years from publication. Supporting documents are available upon request to the corresponding author.

**Conflicts of Interest:** The authors declare no conflict of interest.

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
