# Peer review of "Oral Health Professionals’ and Patients’ Opinions of Type-2 Diabetes Screenings in an Oral Healthcare Setting"

_endocrines, doi:10.3390/endocrines4010005_

Round 1

Reviewer 1 Report

I am a dietitian and have taught dental assistants to look for diabetes in patients they see so I felt this study was valuable. My only comment is the sample was low.

Author Response

  • I am a dietitian and have taught dental assistants to look for diabetes in patients they see so I felt this study was valuable.

Reply: Many thanks for your kind comments.

2) My only comment is the sample was low.

Reply: We agree with the reviewer regarding sample size.  In particular, the completion rate from patients and to some extent OHPs, was low.  This fact is now better acknowledged among the limitations of the study. It was also recognised as a limitation of the statistical analysis.

However, the response rate achieved among OHPs was 50% and representation across oral health professionals was also achieved. The completion rate from patients was lower (35.1%).  Patients not completing our protocol can be considered a limitation of our study. Further research is needed to understand the reasons for this and develop strategies to overcome it.

This has been also clarified in the discussion.

Reviewer 2 Report

Please consider explaining the method section in more detail. The author mentioned that the approach is fully

describe elsewhere but should consider giving the outline. 

Please explain what The Australian Diabetes Risk Assessment Tool is.

Please explain why 22.9% mentioned that it was very difficult to get a response from the general medical practitioners, as mentioned in Table 1.

Please explain why 17.9%  mentioned being slightly dissatisfied with the review of prediabetes and type 2 diabetes by the general medical practitioner, as mentioned in Table 1.

In the discussion section, please discuss the current situation with prediabetes and type 2 diabetes screening with the objective number, if possible and how much impact this study can make. What are potential barriers and any financial implications?

Author Response

Reviewer 2.

  • Please consider explaining the method section in more detail. The author mentioned that the approach is fully describe elsewhere but should consider giving the outline. 

Reply: The materials and methods section described the parts of the model most important to understanding the analysis presented in the manuscript. 

  • Please explain what The Australian Diabetes Risk Assessment Tool is.

Reply: Following this Reviewer’s comment, the Australian Diabetes Risk Assessment Tool is now better described.

  • Please explain why 22.9% mentioned that it was very difficult to get a response from the general medical practitioners, as mentioned in Table 1.
  • Please explain why 17.9%  mentioned being slightly dissatisfied with the review of prediabetes and type 2 diabetes by the general medical practitioner, as mentioned in Table 1.

Reply: OHP participants were asked to include comments to their responses.  However, the majority chose not to do so.  Among those who did include a comment, the major justification for their responses was the general lack of reply to the referral from the GP. However, as explained in the manuscript, it was difficult to identify the reason of this lack of response.

  • In the discussion section, please discuss the current situation with prediabetes and type 2 diabetes screening with the objective number, if possible and how much impact this study can make. What are potential barriers and any financial implications?

Reply: The impact of the study is published in:  Mariño R, Priede A, King M, Adams G, Morgan M. Oral health professionals screening for undiagnosed Type-2 diabetes and pre-diabetes. BMC Endocrine Disorders 2022;22:183. https://doi.org/10.1186/s12902-022-01100-9. Readers were referred to this publication in the manuscript.

Also: Mariño R, Priede A, King M, Lopez D, Adams G, Morgan M. Attitudes and opinions of Oral healthcare professionals on screening for Type-2 diabetes. BMC Health Service Research 2021;21:743. https://doi.org/10.1186/s12913-021-06756-y.

The financial implications are presented in the final report and in another publication.  Please see: Gao L, Tan E, Mariño R, King M, Priede A, Adams G, Sicari M, Moodie M. Long-term cost-effectiveness of a screening intervention to early identify pre-diabetes in the oral healthcare setting. Endocrines. 2022;3: 753-764. https://doi.org/10.3390/endocrines3040062.

Reviewer 3 Report

The current manuscript submitted by Rodrigo M et al and entitled “Oral health professionals’ and patients’ opinions of type-2 diabetes screenings in an oral healthcare setting”. Authors described the comprehensive understanding of the approach to identify patients with prediabetes or T2D, an evaluation was undertaken to investigate OHPs and participating patients’ opinions on the screening program. This information will be used to obtain a better understanding of the acceptability of this approach among OHPs and patients, and subsequently used to inform the development of continuing education programs specifically focused on T2D/pre- diabetes prevention, identification, and management.

I have minor concern:

1) It would be better if authors could provide the data from the patient demography like HbA1c, serum glucose levels, TG and cholesterol, etc.

2) Is these patients have any other comorbidities?

Author Response

Reviewer 3.

The current manuscript submitted by Rodrigo M et al and entitled “Oral health professionals’ and patients’ opinions of type-2 diabetes screenings in an oral healthcare setting”. Authors described the comprehensive understanding of the approach to identify patients with prediabetes or T2D, an evaluation was undertaken to investigate OHPs and participating patients’ opinions on the screening program. This information will be used to obtain a better understanding of the acceptability of this approach among OHPs and patients, and subsequently used to inform the development of continuing education programs specifically focused on T2D/pre- diabetes prevention, identification, and management.

I have minor concern:

  • It would be better if authors could provide the data from the patient demography like HbA1c, serum glucose levels, TG and cholesterol, etc.
  • Is these patients have any other comorbidities?

Reply.  Patients’ demographics (apart from age and sex), and comorbidity data were not collected from those who responded to the completion questionnaire.  While we agree with the importance of this information, the main purpose of the study was to assess an oral healthcare practice-based model that identifies patients with prediabetes or type-2 diabetes using oral health professionals recruited from private rural and urban dental practices in Victoria.